# Incidence and risk factors of kidney impairment on patients with COVID-19: A meta-analysis of 10180 patients

Qixin Yang[1,2]*, Xiyao Yang[1]

**1** National Clinical Research Center for Geriatric Disorders, Xiangya Hospital, Central South University, Changsha, Hunan, China, **2** Department of Cardiovascular Medicine, Xiangya Hospital, Central South University, Changsha, Hunan, China

* yangqixin@csu.edu.cn

**Data Availability Statement:** All relevant data are within the manuscript and its Supporting Information files.

**Funding:** Q.Y is supported by Chinese Government Scholarship (University Graduate Program) in

## Abstract

### Background

The novel coronavirus is pandemic around the world. Several researchers have given the evidence of impacts of COVID-19 on the respiratory, cardiovascular and gastrointestinal system. Studies still have debated on kidney injury of COVID-19 patients. The purpose of the meta-analysis was to evaluate the association of kidney impairment with the development of COVID-19.

### Methods

The PubMed, Embase and MedRxiv databases were searched until May 1, 2020. We extracted data from eligible studies to summarize the clinical manifestations and laboratory indexes of kidney injury on COVID-19 infection patients and further compared the prevalence of acute kidney injury (AKI) and the mean differences of three biomarkers between in ICU/severe and non-ICU/non-severe cases. Heterogeneity was evaluated using the $I^2$ method.

### Results

In the sum of 24 studies with 10180 patients were included in this analysis. The pooled prevalence of AKI, increased serum creatinine (Scr), increased blood urea nitrogen (BUN), increased D-dimer, proteinuria and hematuria in patients with COVID-19 were 16.2%, 8.3%, 6.2%, 49.8%, 50.1% and 30.3% respectively. Moreover, the means of Scr, BUN and D-dimer were shown 6.4-folds, 1.8-folds and 0.67-folds, respectively, higher in ICU/severe cases than in corresponding non-ICU/non-severe patients. The prevalence of AKI was about 30 folds higher in ICU/severe patients compared with the non-ICU/non-severe cases.

### Conclusions

Overall, we assessed the incidences of the clinic and laboratory features of kidney injury in COVID-19 patients. And kidney dysfunction may be a risk factor for COVID-19 patients developing into the severe condition. In reverse, COVID-19 can also cause damage to the kidney.

Central South University with grant number 31801-160170002. The funders had no role in study design, data collection and analysis, decision to publish, or preparation of the manuscript.

**Competing interests:** The authors have declared that no competing interests exist.

## Introduction

In December 2019, a group of pneumonia cases caused by an unknown virus was first reported in Wuhan, Hubei province, China [1, 2]. Those cases have similar symptoms of virus infection, including fever, fatigue, and dry cough as well as myalgia, dyspnea [1, 2]. WHO has officially named this novel coronavirus as severe acute respiratory syndrome coronavirus 2 (SARS-CoV-2) after the pathogen was isolated and identified [3, 4]. Nowadays, this novel coronavirus is causing COVID-19 epidemic on the international scale due to its highly transmissive and contagiousness, compared with other coronavirus infection diseases including Middle East Respiratory Syndrome (MERS) and Severe Acute Respiratory Syndrome (SARS) [5]. As of October 15, 2020, a total of 38129806 confirmed cases involved in 185 countries and regions, and the numbers continue to rise. And SARS-CoV-2 mainly causes a series of the clinical characteristics in the respiratory system, such as asymptomatic infection, mild upper respiratory tract illness, severe acute respiratory distress syndrome, respiratory failure and even death [1, 2, 6].

The pathogenic mechanism of SARS-CoV-2 is binding to membrane ACE2 for entering into pulmonary cells [4]. And ACE2 is widely distributing in several vital organs including lung, heart, kidney and intestine [7]. Apart from the respiratory symptom, SARS-CoV-2 also caused cardiovascular damage, not only led to acute cardiac injury (ACI) with an increased high-sensitivity cardiac troponin I (hs-cTnI) in clinic [1]. On the other hands, patients with pre-existing cardiovascular diseases (CVDs) are more likely developed into the severe condition and even contribute to highly mortality [1, 8, 9]. Moreover, SARS-CoV-2 has an impact on the gastrointestinal system, bringing symptom like diarrhea with a statistically significant difference, which may be underestimated on clinical diagnosis [10]. Further study has proved that SARS-CoV-2 infects the gastrointestinal tract, the results of histologic and immunofluorescent staining of gastrointestinal tissues from COVID-19 patients were showed that the existence of ACE2 receptor and viral nucleocapsid protein in the cytoplasm of gastric, duodenal, and rectum glandular epithelial cell [11].

Therefore, we are also concerned whether SARS-CoV-2 causes kidney dysfunction and whether COVID-19 patients with kidney impairment are at a higher risk. Some clinical studies have focus attention on kidney injury of COVID-19 patients. Zhen Li et al. has shown that a large proportion of COVID-19 patients is accompanied by kidney dysfunction, including proteinuria, hematuria, increased serum creatine and blood urea nitrogen [12]. Yichun Cheng also demonstrated that kidney injury is associated with in-hospital death of COVID-19 patients [13]. However, Luwen Wang thought that SARS-CoV-2 did not cause obviously kidney damage to patients [14]. With issue arising, a meta-analysis with large clinical samples is desperately warranted to produce a convincible result.

## Methods

Our study was conducted following Preferred Reporting Items for Systematic Reviews and Meta-Analyses of individual participant data (the PRISMA-IPD) statement [15].

### Data source, search strategy and selection criteria

PubMed, EMBASE, and MedRxiv databases were applied for searching studies published from December 2019 to May 2020. To identify all the articles displaying the renal injury and impairment in COVID -19, we used the following keywords or search terms alone and in combination: "novel coronavirus", "China", "HCoV", "nCoV" "Wuhan", "COVID-19" "SARS-CoV-2", "clinical", "laboratory", "kidney", "Acute Kidney Injury", "proteinuria" and "hematuria". Detailed search strategies were illustrated in **Fig 1**.

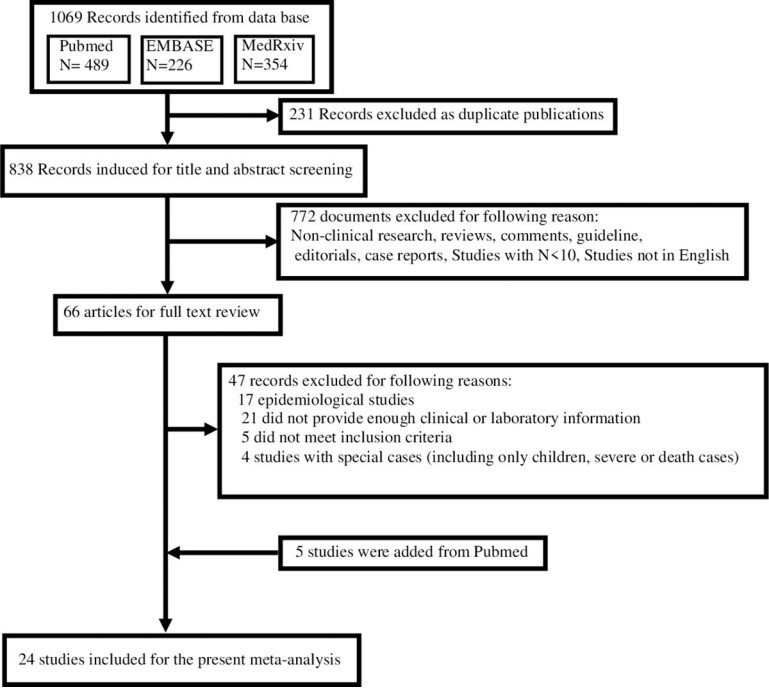

**Fig 1. PRISMA flowchart of the study selection process.**

Two reviewers (Q.Y. & X.Y.) independently screened the titles, abstracts and followed the full texts to decided which studies should be included in. Inclusion criteria are as follows:(1) comparative studies: randomized controlled trials RCTs or non-RCTs published only restricted in English; (2) patients in the studies should be confirmed to have been infected by COVID-19; (3)studies containing information about the clinical or laboratory characteristics (4) studies containing the comorbidities of kidney dysfunction and the outcome of kidney impairment. Exclusion criteria are (1) studies that less than 10 patients were included; (2) case reports, editorials, comments, non-clinical studies, reviews, studies without reliable information; (3) studies with special populations (e.g., only focused on children or severe or death cases).

## Data extraction and study quality assessment

Prevalence of comorbidities and clinical manifestations of kidney damage, including AKI, proteinuria and hematuria, together with laboratory indexes of kidney impairment (confirmed by elevation of Scr, BUN and D-dimer) were extracted from the identified studies. The subgroup measure parameters were to compare the incidences of AKI and the mean differences of the three laboratory indicators among ICU and Non-ICU cases (severe and non-severe data as the second choice if ICU data was not provided). Cochrane Collaboration's tool was applied to evaluate the risk of bias.

## Statistical analysis

All analyses were performed using OpenMeta Analyst (version 12.11.14) (http://www.cebm.brown.edu/openmeta/) and Review Manager (version 5.3). Forest plots were used to depict the incidences of clinical and laboratory features of kidney dysfunction of COVID-19 patients. The odds ratio (OR, 95% confidence intervals (CI)) and mean differences (MD, 95%

confidence intervals (CI)) were used to illustrate the comprehensive effects of COVID-19 occurring in ICU/severe patients and non-ICU/non-severe patients. And $I^2$ statistics were used to assess the statistical heterogeneity. The fixed-effect model was used if $I^2 < 50\%$ and the random effect model was used if $I^2 \geq 50\%$ [8]. The funnel plots were used to show the risk of publication bias.

## Results

### Selected literature and studies characteristics

At initially, we have searched a total of 838 studies after 231 duplicate studies were identified. Following reviewed the titles and abstracts, we ruled out 772 non-clinical research, reviews, comments, case reports and studies of participants less than 10. With the remaining 66 documents, we reviewed and evaluated the whole articles with detailed information. We further excluded 47 records for multiple reasons such as lacking enough clinical information and only demonstrated the exceptional cases. In the result, we identified 24 eligible studies meeting our inclusion criteria for our meta-analysis, including 10180 COVID-19 positive patients (**Fig 1**). All of them were retrospective, descriptive observational studies including 17 single-center and 7 multiple-center studies from different countries and regions, which were mainly conducted between December 2019 and May 2020.

The epidemiological and clinical characteristics of COVID-19 from 19 included studies were illustrated in **Table 1**. And we also described the prevalence of the complications of kidney injury on clinic and laboratory features. Among all selected studies, the infected men accounted for a more substantial proportion than women and the men to women ratio was 1.4. The mean age of the participants was 54.6 years (95% CI, 51.2–58.0).

### AKI and biomarkers of kidney dysfunction

Our outcome of meta-analysis for identified studies suggested that the AKI occurred 16.2% (95% CI 7.0–25.3%) in COVID-19 patients (**Fig 2A**). According to the Kidney Disease: Improving Global Outcomes (KDIGO) guidelines [16] and the limited clinical and laboratory information acquired from those studies, we used several indicators to display the comorbidities of kidney dysfunction. And the most prevalent laboratory indexes were increased Scr (8.3%, 95% CI 4.3–12.3%) (**Fig 2B**), increased BUN (6.2%, 95% CI 2.4–10.1%) (**Fig 2C**) and increased D-dimer (49.8%, 95% CI 35.4–64.2%) (**Fig 2D**). We included D-dimer for the reason that the elimination of D-dimer protein partially happens through kidney and high D-dimer is associated with the dysfunction of kidney [17]. However, the $I^2$ index ranging from 82% to 99% revealed significant heterogeneity in the evaluation of AKI, Scr, BUN and D-dimer among the included studies (P<0.001) (**Fig 2A** to **2D**).

### Risk stratification factors for COVID-19

To identify the risk factors for critical illnesses of COVID-19 patients, we then analyzed the relevance of the AKI and the three laboratory indexes with the clinical severity through comparing the incidences of AKI and mean differences of those biomarkers between ICU/severe and non-ICU/non-severe cases. Following the results of the heterogeneity test were all shown as $I^2$ <50%, we applied the fixed-effect model for further investigations. For AKI, the result from 9 studies including 7313 patients showed that the AKI occurred statistically significantly higher in ICU cases (73.2%) compared with non-ICU cases (16.5%) [OR 29.51 95%CI (24.45, 35.62), Z = 35.27, P<0.00001] (**Fig 2A**). In terms of laboratory results, there were considerable differences between ICU and non-ICU cases in Scr (MD = 6.38 μmol/L, 95%CI 3.10–9.65, 13

**Table 1.  Study characteristics including number, location, age, sex and kidney impairment of patients of the 24 included studies.**

| Studies | Study period | Number of patients | location | Mean age (SD) | Sex (male,%) | Kidney metabolic diseases | | | | | |
|---|---|---|---|---|---|---|---|---|---|---|---|
| | | | | | | AKI % | Increased Scr % | Increased BUN % | Increased D-dimer % | Proteinuria % | Hematuria % |
| Zhen Li et al. [12] | 1/6/2020-2/21/2020 | 193 | Wuhan, Huangshi, Chongqing | 56.7 (15.6) | 95(49%) | 28.5 | 10.4 | 14.0 | 58.8 | 58.9 | 44.2 |
| Yichun Cheng et al. [13] | 1/28/2020-2/11/2020 | 710 | Wuhan | 61.7 (14.8) | 374 (52.7%) | 3.2 | 15.5 | - | 77.7 | 44.0 | 26.9 |
| Haifeng Zhou et al. [37] | 2/2/2020-2/29/2020 | 178 | Wuhan | 47.7 (19.3) | 72 (40.4%) | - | 0.0 | 2.8 | - | 34.9 | 28.9 |
| Fei Zhou et al. [38] | 12/29/2019-1/31/2020 | 191 | Wuhan | 56.3 (15.6) | 119 (62%) | 14.7 | 4.3 | - | 68.0 | - | - |
| Nanshan Chen et al. [6] | 1/1/2020-1/20/2020 | 99 | Wuhan | 55.5 (13.1) | 67(68%) | 3.0 | - | 6.1 | 36.4 | - | - |
| Chaolin Huang et al. [1] | 12/16/2019-1/2/2020 | 41 | Wuhan | 49.3 (12.6) | 30(73%) | 7.3 | 9.8 | - | - | - | - |
| Qingxian Cai et al. [39] | 1/11/2020-2/6/2020 | 298 | Shenzhen | 47.0 (20.7) | 149 (50%) | 5.7 | 4.4 | 4.0 | 35.7 | - | - |
| Weijie Guan et al. [40] | 12/11/2019-1/29/2020 | 1099 | Nationwide | 46.7 (17.0) | 637 (58%) | 0.5 | 1.6 | - | 46.4 | - | |
| Dawei Wang et al. [2] | 1/1/2020-1/28/2020 | 138 | Wuhan | 55.3 (19.3) | 75 (54.3%) | 3.6 | - | - | - | - | - |
| Jiatao Lu et al. [41] | 1/21/2020-2/5/2020 | 577 | Wuhan | 53.3(20) | 254 (44%) | - | 3.0 | - | 33.5 | - | - |
| Jinjin Zhang et al. [25] | 1/16/2020-2/3/2020 | 140 | Wuhan | 57.2 (14.8) | 71 (50.7%) | - | - | - | 43.2 | - | - |
| Weihe Quan et al. [42] | 2/25/2020-3/13/2020 | 18 | Shenzhen, Wuhan | 60.3 (15.3) | NA | - | - | - | - | 22.2 | 16.7 |
| Yonghao Xu et al. [43] | 2/28/2020 | 45 | Guangdong | 56.7 (15.4) | 29 (64.4%) | 15.6 | - | - | - | - | - |
| Guang Chen et al. [44] | 12/19/2019-1/27/2020 | 21 | Wuhan | 57.0 (11.1) | 17(81%) | - | - | - | - | - | - |
| Yanli Liu et al. [45] | 1/2/2020-2/1/2020 | 109 | Wuhan | 54.7 (17.0) | 59 (54.1%) | - | - | - | - | - | - |
| Jingyuan Liu et al. [46] | 1/13/2020-1/31/2020 | 61 | Beijing | 42.3 (15.7) | 31 (50.8%) | - | - | - | - | - | - |
| Lei Liu et al. [47] | 1/11/2020-2/6/2020 | 51 | Chongqing | 43.3 (12.6) | 32(63%) | - | - | - | - | - | - |
| Zhichao Feng et al. [48] | 1/17/2020-2/1/2020 | 141 | Hunan | 44.3 (15.3) | 72 (51.1%) | - | - | - | - | - | - |
| Hongzhou Lu et al. [49] | 1/20/20-2/19/2020 | 265 | Shanghai | NA | NA | - | - | - | - | - | - |
| Saurabh Aggarwal et al. [50] | 3/1/2020-4/4/2020 | 43 | Jersey, USA | 66.7 (42.2) | 32 (74.4%) | 68.8 | 68.8 | - | - | - | - |
| Spinello Antinori et al. [51] | 2/23/2020-3/20/2020 | 35 | Italy | 61.0 (13.3) | 26 (74.3%) | 22.8 | - | - | - | - | - |
| Matthew J Cummings et al. [52] | 3/2/2020-4/1/2020 | 257 | New York, USA | 61.7 (15.6) | 171 (66.5%) | - | - | - | - | 0.9 | - |

*(Continued)*

**Table 1.** (Continued)

| Studies | Study period | Number of patients | location | Mean age (SD) | Sex (male,%) | Kidney metabolic diseases | | | | | |
|---|---|---|---|---|---|---|---|---|---|---|---|
| | | | | | | AKI % | Increased Scr % | Increased BUN % | Increased D-dimer % | Proteinuria % | Hematuria % |
| Matt Arentz et al. [53] | NA | 21 | Seattle, USA | 68.3 (36.3) | 11 (52.4%) | 19.1 | - | - | - | - | - |
| Jamie S. Hirsch et al. [24] | 3/1/2020-4/5/2020 | 5449 | New York, USA | 63.7 (17.0) | 3317 (60.9%) | 36.6 | - | - | - | - | - |

AKI, acute kidney injury; Scr serum creatinine; BUN blood urea nitrogen.

studies, n = 1267) (**Fig 2B**), BUN(MD = 1.84μmol/L, 95%CI 1.44–2.25, 7 studies, n = 701) (**Fig 2C**) and D-dimer (MD = 0.67mg/L, 95%CI 0.54–0.79, 12 studies, n = 1553) (**Fig 2D**). In conclusion, AKI, increased Scr, BUN and D-dimer were prominent features when patients developed into critical conditions (all P<0.001) (**Fig 2A** to **2D**).

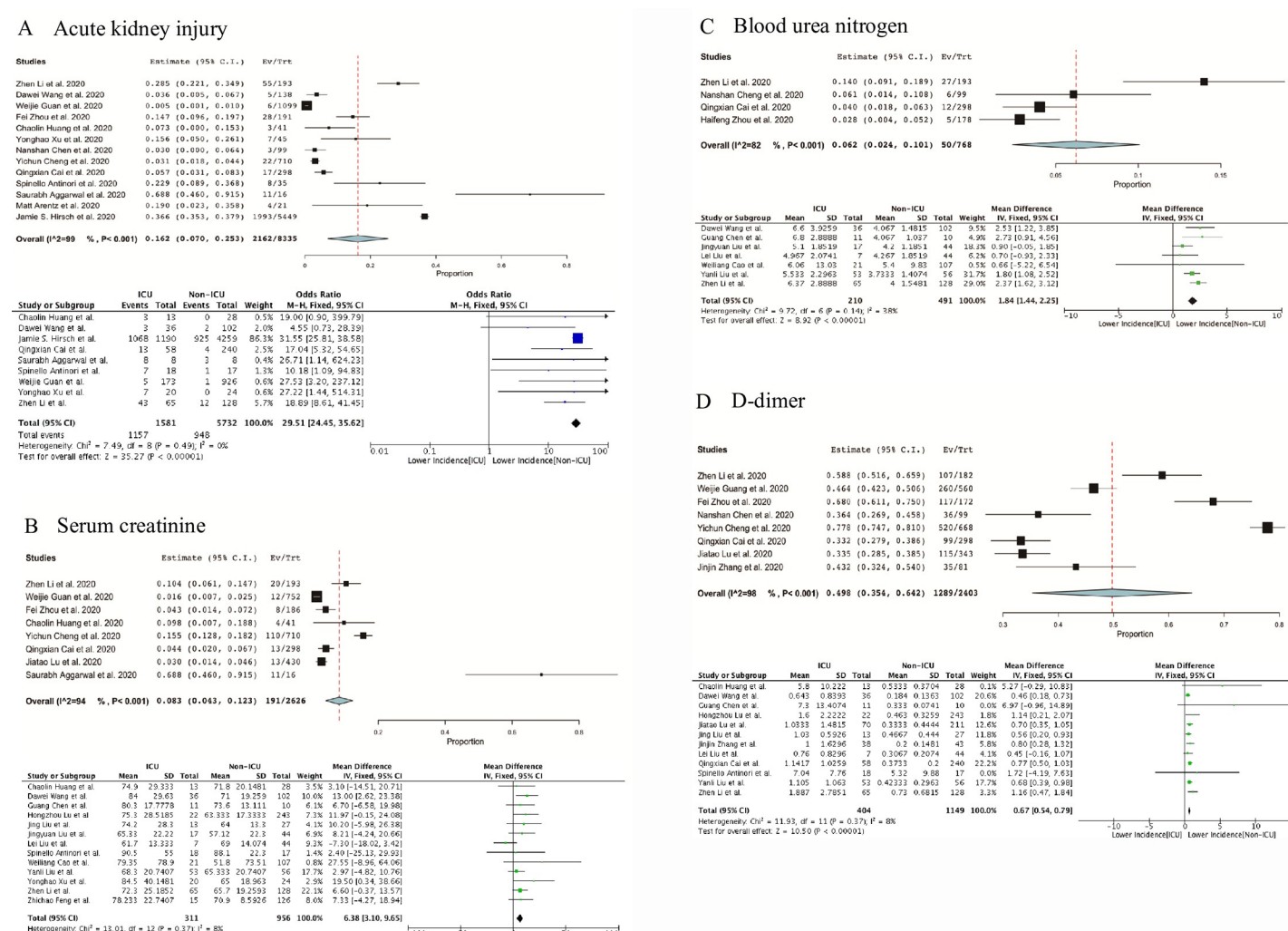

**Fig 2. Meta-analysis of incidences of AKI and three laboratory indexes of kidney injury.** Forest plots represent the comparisons of the prevalence of AKI and mean differences (MD) in three laboratory indicators between ICU/severe and non-ICU/non-severe cases. A, serum creatinine (Scr, μmol/L); B, blood urea nitrogen (BUN, μmol/L); C, D-dimer(mg/L).

A  Proteinuria

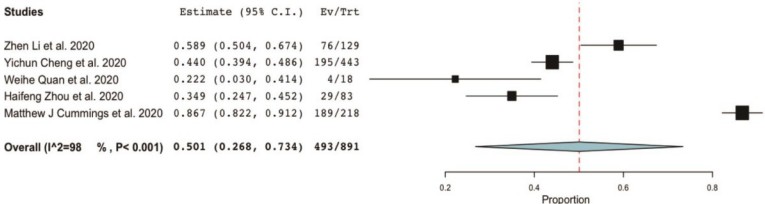

B  Hematuria

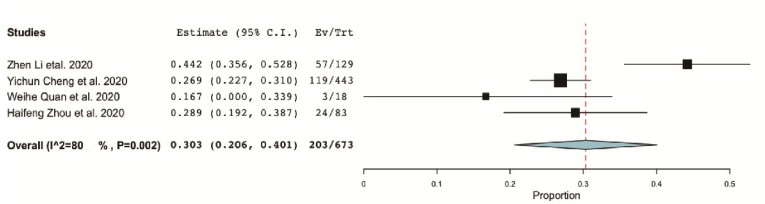

**Fig 3. Meta-analysis of incidences of two clinical characteristics of kidney injury.** A, proteinuria; B, hematuria.

## Clinical characteristics of kidney impairment

AKI is a risk factor of proteinuria and subsequently can be developed into chronic kidney disease(CKD) [18]. Here, we sought to further explore the clinical effects of kidney impairment caused by COVID-19, and we analyzed another two clinical features among COVID-19 patients. The results show that the most prevalent of kidney injury comorbidities were proteinuria (50.1%, 95%CI 26.8%-73.4%) (**Fig 3A**) and hematuria (30.3%, 95%CI 20.6%-40.1%) (**Fig 3B**) with high heterogeneity (both $I^2 > 80$%).

## Sensitivity analysis and bias assessment

In the end, the funnel plots displayed symmetrical distributions of the effect sizes of AKI, Scr, BUN and D-dimer, and presented no obvious publication bias (**Fig 4A to 4D**).

## Discussion

The COVID-19 has affected hundreds of millions of people posing a huge healthy threaten and bring a major burden to public healthcare institutions around the world. Compared with the other two pathogenic coronaviruses family members SARS-CoV and MERS-CoV, SARS-CoV-2 is higher contagious causing global pandemic, whereas each of which has its own clinical manifestation. Studies have been reported that SARS-CoV-2 is sharing highly 79.6% genome sequence identity as well as high molecular structure similarity with SARS-CoV [4, 19]. Therefore, SARS-CoV-2 uses ACE2 as a cellular entry receptor as SARS-CoV [4, 20]. ACE2 is highly expressed in multiple systems and tissues, mainly in the respiratory, cardiovascular, renal and gastrointestinal systems [7]. In addition to respiratory diseases and cardiac damage caused by SARS-CoV-2 through ACE2, we still need to consider the possibility of kidney effects on COVID-19 patients.

The meta-analysis was based on data from 24 studies with confirmed 10180 COVID-19 cases in worldwide. In all cases, men were a more significant population around 58% than

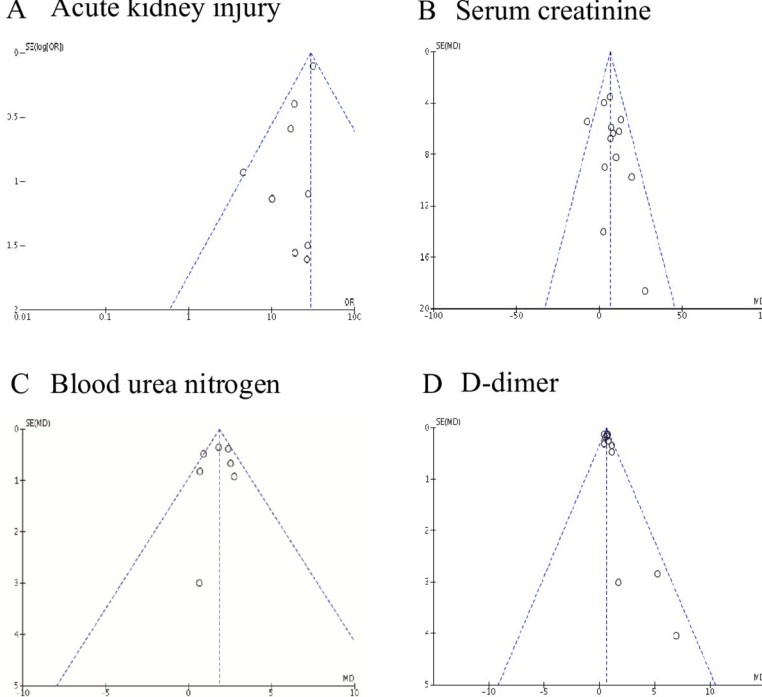

**Fig 4. Funnel plots of the comparisons of AKI.** (A), Scr (B), BUN (C) and D-dimer (D) between ICU/severe and non-ICU/severe patients.

women, which has similar infection characteristics as MERS and SARS [21, 22]. Our meta-analysis has shown that the prevalence of AKI is approximately 16.2%, and other laboratory biomarkers reflecting renal injuries such as increased Scr, BUN and D-dimer were presented in 8.3%, 6.2%, 49.8%, respectively. Moreover, the clinical features of kidney dysfunction are even higher than cardiovascular diseases in COVID-19 patients, proteinuria is 50.1% and hematuria is 30.3%, while hypertension and diabetes were showed around 8% and 5% in Jing Yang's study [9]. And other researchers have showed the similar results of the prevalence of increased Scr(9.6%), BUN(13.7%) and proteinuria(57.2%) in their meta analysis [23].

We have further analyzed the correlation ship between the COVID-19 and kidney dysfunction while other studies have not noted. When compared the ICU/severe and non-ICU/non-severe cases, our results demonstrated that the AKI happened 30-folds higher in critical condition, while the incidence of acute cardiac injury was around 13-folds higher in severe disease in Bo Li's study [8]. Moreover, the indexes of Scr, BUN and D-dimer were demonstrated 6.4-folds, 1.8-folds and 0.67-folds, respectively, higher when patients developed into ICU/severe illness. At all, the AKI is highly associated with severe COVID-19 and more susceptibility than cardiac damage so that we should pay more attention to protecting the normal function and recovery of kidney in clinic.

However, there are still some limitations for this meta-analysis. Firstly, due to the restriction of clinic information from the literatures and most of the studies did not include in the death cases or the mortality of COVID-19, the association between kidney impairment and COVID-19-induced death was not be analyzed in our meta-analysis. And we hardly could include study compared the complications of kidney injury between ICU/severe and non-ICU/non-severe patients. In which cases, we did not perform sensitivity analysis and subgroup analysis for proteinuria, hematuria or uric acid. Secondly, we found that the high statistic

heterogeneity in the prevalence of kidney injury analysis. The reasons are related to the study designs and significant variations among studies in the sample sizes. Thirdly, therapies under investigation for COVID-19 may have kidney side effects as lots of drugs are nephrotoxic such as aminoglycosides, ACE inhibitors and nonsteroidal anti-inflammatory drugs (NSAIDs), we are not sure whether some clinical data we got have such possibility involved, and we could not rule out the influences caused by drugs on kidney during the hospitalization.

In this meta-analysis, we have showed AKI is a critical bio-indicator for COVID-19 patients developed into severe condition. Other clinical researchers founded that black race was more susceptible to AKI [24]. Also, there are several risk factors were reported to have worse outcomes even higher mortality when infected with COVID-19, including increased age, higher BMI, smoking, male sex, African Americans and Hispanics ethnicity, and people with other comorbidities, especially cardiovascular diseases [25–28]. As a result, risk stratification, individualize interventions and early therapeutic measures are required for clinical management in order to prevent the progression of AKI and reduce mortality and morbidity.

Some mechanisms are involved for explanation of kidney injury during the COVID-19 infection episodes. Firstly, ACE-2 distributes on tubular epithelial cells of the kidney with a higher expression level compared to the lung. Thus, the kidney is also a direct aim organ attacked by SARS-CoV-2 entering into target cells through ACE-2 acting as the way in the lung. In most recently pathological autopsy results, researchers found that coronavirus-like particles were directly discovered in kidney through immunohistochemistry on patients died with COVID-19 and SARS-CoV nucleoprotein antibody were indirectly detected in kidney tissue by immunofluorescent staining on COVID-19 patients [29, 30]. All those evidences have been demonstrated that the infection of COVID-19 and immunodysregulation on kidney, which caused AKI in the early stage and significant comorbidities such as hypertension, chronic kidney disease in the long term [30]. Moreover, coronavirus-like particles also were detected in other critical organs including respiratory system and gastrointestinal tract, causing obstructive sleep apnoea, diabetes and obesity [30]. Therefore, we still need to follow-up the outcome of those COVID-19 patients in the future. And establishing the chronic diseases community will play a critical role in the management and treatment of patients affected by this epidemic disease.

Besides, the crosstalk relationship between lung and kidney. Kidney damage can be caused by circulating inflammatory factors such as tumor necrosis factor (TNF)-α and interleukin (IL)-6, which are originated from pneumonia, happened in the lung. Furthermore, the local inflammatory response from injury and death renal cells will accelerate damage in the development of AKI as well as other organs [31, 32]. Thus, to reduce the possibility of developing into critically illness and the mortality risk for COVID-19 patients, applying more protective measures and supportive medication interventions is necessary, which has a significant influence for the kidney care of patients, including the application of drugs with mild kidney adverse effects, renal replacement therapies (RRT) like blood filtering and purification treatments etc. For example, continuous RRT with continuous veno-venous haemo-dialysis modality (CVVHD) is an efficacious way to prevent the progression of severity in COVID-19 patients, especially those with refractory hypoxaemia and unstable haemodynamic status [33]. Also, the application of CVVHD is helpful to increase cytokine removal which mitigates the kidney damage induced by inflammatory factors [34]. Moreover, RRT coupled with low-flow extracorporeal carbon dioxide removal (ECCO$_2$R) would be a better and supportive therapy for the critical illness, however, still need to be verification with sufficiently clinical trials [33].

On the other hand, we should analyze the reasons accounting for underestimating kidney impairment in COVID-19 patients in clinic. Firstly, the laboratory tests of blood chemistry analysis, including Scr and BUN, will only elevate into abnormal range when kidney lost at

least 50% function because of the potent compensatory ability of kidney. From our results, we also found that the proportions of aberrant urinalysis were more than the percentage of increased plasma biomarkers. Secondly, the difficulty of precise diagnosis of AKI is another possible aspect responsible for the underrating of AKI. The detection rate of AKI mainly depends on the fluctuation of Scr and the frequency of Scr testing. And a higher incidence of AKI will be detected with adjusted denser Scr testing frequency. Therefore, more accurate strategies should be applied to the clinic when considered AKI [35]. There are several bio-markers can be used for monitoring the kidney function such as the level fluctuation of creati-nine and urine output with the volume and hemodynamic status, and some novel indicators also should be added in for precisely stratifying the AKI severity such asTIMP-2 (tissue inhibi-tor of metalloproteinase 2 (TIMP-2) and insulin-like growth factor binding protein 7(IGFBP7) [36].

In conclusion, SARS-CoV-2 causes renal injury progressing to severe AKI. At the same time, AKI is a life-threatening complication associating with a severer condition in COVID-19 patients. Therefore, we should focus more attention to kidney damage at the early stage when patients are confirmed been infected by COVID-19 according to a series of accurate clinical parameters. And more protective therapies are urged to be confirmed the clinical value on COVID-19 patients with kidney dysfunction.

## Supporting information

**S1 Checklist. PRISMA 2009 checklist.**
(DOC)

## Author Contributions

**Conceptualization:** Xiyao Yang.

**Data curation:** Qixin Yang, Xiyao Yang.

**Formal analysis:** Qixin Yang, Xiyao Yang.

**Funding acquisition:** Qixin Yang.

**Investigation:** Qixin Yang, Xiyao Yang.

**Methodology:** Qixin Yang.

**Project administration:** Qixin Yang.

**Resources:** Qixin Yang.

**Software:** Qixin Yang.

**Supervision:** Qixin Yang.

**Validation:** Qixin Yang.

**Visualization:** Qixin Yang.

**Writing – original draft:** Qixin Yang.

**Writing – review & editing:** Qixin Yang.

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
