## [Decision Letter · Decision Letter 0]

17 Sep 2020

PONE-D-20-15148

Incidence and risk factors of kidney impairment on patients with COVID-19: a systematic review and meta-analysis

PLOS ONE

Dear Dr. Yang,

Thank you for submitting your manuscript to PLOS ONE. After careful consideration, we feel that it has merit but does not fully meet PLOS ONE’s publication criteria as it currently stands. Therefore, we invite you to submit a revised version of the manuscript that addresses the points raised during the review process.

We look forward to receiving your revised manuscript.

Kind regards,

Chiara Lazzeri

Academic Editor

PLOS ONE

Journal Requirements:

Reviewers' comments:

Reviewer's Responses to Questions

**Comments to the Author**

1. Is the manuscript technically sound, and do the data support the conclusions?

Reviewer #1: Yes

Reviewer #2: Partly

2. Has the statistical analysis been performed appropriately and rigorously? 

Reviewer #1: Yes

Reviewer #2: Yes

3. Have the authors made all data underlying the findings in their manuscript fully available?

Reviewer #1: Yes

Reviewer #2: Yes

4. Is the manuscript presented in an intelligible fashion and written in standard English?

Reviewer #1: Yes

Reviewer #2: No

5. Review Comments to the Author

Reviewer #1: 1. The studies included was limited in China.To get a more comprehensive understanding of kidney impairment on patients with COVID-19, more researches from different countries and regions will be better. 

2. Recent study ( such as Crit Care. 2020 Jun 18;24(1):356,and probably more)  has reported the acute renal impairment on COVID-19.The manuscript should compare with  similar systematic reviews in discussion.

3. It is not mentioned whether more than one investigators work independently to decide which studies should be included.

Reviewer #2: This is a timely review paper on COVID and kidney disease. The authors start with 838 studies, down to 66 then 19. They then summarize the clinical characteristics of the patient with AKI, the incidence if in the ICU or out, and associated proteinuria and hematuria as risk factors.

Comments

1. The current study does not add to the current major questions about COVID and the kidney. What is going on in biopsies? What are the main comorbidities? Who will survive and who will not? What dialysis modalities are the best for those critically ill? How does organ dysfunction in other organs contribute?

2. There are many other key original papers on COVID and kidney disease that are not discussed and reflect very important clinical data

6. PLOS authors have the option to publish the peer review history of their article (what does this mean?). If published, this will include your full peer review and any attached files.

Reviewer #1: No

Reviewer #2: No

---

## [Author Response · Author response to Decision Letter 0]

16 Oct 2020

Dear Editor Chiara Lazzeri,

Many thanks for proving us an opportunity for us to make further revisions for our manuscript titled “Incidence and risk factors of kidney impairment on patients with COVID-19: a systematic review and meta-analysis” submitted to Plos One. We appreciate the patience and efforts that you and the reviewers have worked on our paper, and more importantly, provided many thoughtful suggestions and valuable comments for our research study. We have been able to address most of the questions raised from the reviewers. We have made traceable changes and highlight the revisions in our manuscript.

Here is the point-by-point response to the reviewers’ comments and concerns.

Reviewer #1: 1. The studies included was limited in China. To get a more comprehensive understanding of kidney impairment on patients with COVID-19, more researches from different countries and regions will be better. 

2. Recent study ( such as Crit Care. 2020 Jun 18;24(1):356,and probably more) has reported the acute renal impairment on COVID-19.The manuscript should compare with similar systematic reviews in discussion.

3. It is not mentioned whether more than one investigators work independently to decide which studies should be included.

Response: Thank you so much for the first crucial point that we need to include more studies rather than just narrowed in China. Therefore, we have added in more 5 studies from other countries, and accordingly, we have made revisions on our results which can be found in abstract part(page2), result part(page 6-8), and in table1, Figure 1-4[1][2][3][4][5]:

In the sum of 24 studies with 10180 patients were included in this analysis. The pooled prevalence of AKI, increased serum creatinine (Scr), increased blood urea nitrogen (BUN), increased D-dimer, proteinuria and hematuria in patients with COVID-19 were 16.2%, 8.3%, 6.2%, 49.8%, 50.1% and 30.3% respectively. Moreover, the means of Scr, BUN and D-dimer were shown 6.4-folds, 1.8-folds and 0.67-folds, respectively, higher in ICU/severe cases than in corresponding non-ICU/non-severe patients. The prevalence of AKI was about 30 folds higher in ICU/severe patients compared with the non-ICU/non-severe cases.

For the secondly point, we also have made comparisons with other similar meta-analysis, such as Xianghong Yang et al. have reported the similar results of the prevalence of increased Scr(9.6%), BUN(13.7%) and proteinuria(57.2%) in their meta-analysis, which can be found in discussion part(page 9).

For the third point, thanks for reminding us to make more clear statement in our method. We have noted that two reviewers (Q.Y. & X.Y.) independently screened the titles, abstracts and followed the full texts to decided which studies should be included in, which can be found in method part (page 5).

Reviewer #2: This is a timely review paper on COVID and kidney disease. The authors start with 838 studies, down to 66 then 19. They then summarize the clinical characteristics of the patient with AKI, the incidence if in the ICU or out, and associated proteinuria and hematuria as risk factors.

Comments

1. The current study does not add to the current major questions about COVID and the kidney. What is going on in biopsies? What are the main comorbidities? Who will survive and who will not? What dialysis modalities are the best for those critically ill? How does organ dysfunction in other organs contribute?

2. There are many other key original papers on COVID and kidney disease that are not discussed and reflect very important clinical data

Response: Appreciate for those insightful questions that will make our paper, especially discussion part, more comprehensive and significant. Here we will answer the questions point by point:

What is going on in biopsies? What are the main comorbidities?

Answer can be found in discussion part(page11): In most recently pathological autopsy results, researchers found that coronavirus-like particles were directly discovered in kidney through immunohistochemistry on patients died with COVID-19 and SARS-CoV nucleoprotein antibody were indirectly detected in kidney tissue by immunofluorescent staining on COVID-19 patients[6][7]. All those evidences have been demonstrated that the infection of COVID-19 and immunodysregulation on kidney, which caused AKI in the early stage and significant comorbidities such as hypertension, chronic kidney disease in the long term[7]. Moreover, coronavirus-like particles also were detected in other critical organs including respiratory system and gastrointestinal tract, causing obstructive sleep apnoea, diabetes and obesity[7]. 

How does organ dysfunction in other organs contribute?

Answer can be found in discussion part(page12): Firstly, coronavirus-like particles also were detected in other critical organs including respiratory system and gastrointestinal tract, causing obstructive sleep apnoea, diabetes and obesity, which means other organs could be attacked by virus directly [7]. Also, the crosstalk relationship between lung and kidney. Kidney damage can be caused by circulating inflammatory factors such as tumor necrosis factor (TNF)-α and interleukin (IL)-6, which are originated from pneumonia, happened in the lung. Furthermore, the local inflammatory response from injury and death renal cells will accelerate damage in the development of AKI as well as other organs[8][9].

Who will survive and who will not?

Answer can be found in discussion part(page10-11): In this meta-analysis, we have showed AKI is a critical bio-indicator for COVID-19 patients developed into severe condition. Other clinical researchers founded that black race was more susceptible to AKI[5]. Also, there are several risk factors were reported to have worse outcomes even higher mortality when infected with COVID-19, including increased age, higher BMI, smoking, male sex, African Americans and Hispanics ethnicity, and people with other comorbidities, especially cardiovascular diseases[10][11][12][13]. As a result, risk stratification, individualize interventions and early therapeutic measures are required for clinical management in order to prevent the progression of AKI and reduce mortality and morbidity.

What dialysis modalities are the best for those critically ill?

Answer can be found in discussion part(page12): To reduce the possibility of developing into critically illness and the mortality risk for COVID-19 patients, applying more protective measures and supportive medication interventions is necessary, which has a significant influence for the kidney care of patients, including the application of drugs with mild kidney adverse effects, renal replacement therapies (RRT) like blood filtering and purification treatments etc. For example, continuous RRT with continuous veno-venous haemo-dialysis modality (CVVHD) is an efficacious way to prevent the progression of severity in COVID-19 patients, especially those with refractory hypoxaemia and unstable haemodynamic status[14]. Also, the application of CVVHD is helpful to increase cytokine removal which mitigates the kidney damage induced by inflammatory factors[15]. Moreover, RRT coupled with low-flow extracorporeal carbon dioxide removal (ECCO2R) would be a better and supportive therapy for the critical illness, however, still need to be verification with sufficiently clinical trials[14].

There are many other key original papers on COVID and kidney disease that are not discussed and reflect very important clinical data

Many thanks for your thought-provoking point. To address this problem, we have included in more 5 studies from other countries including the original clinic paper especially about the kidney injury and COVID-19, and accordingly, we have made revision on our results which can be found in abstract party(page2), result part(page 6-8), and in table1, Figure 1-4[1][2][3][4][5]:

In the sum of 24 studies with 10180 patients were included in this analysis. The pooled prevalence of AKI, increased serum creatinine (Scr), increased blood urea nitrogen (BUN), increased D-dimer, proteinuria and hematuria in patients with COVID-19 were 16.2%, 8.3%, 6.2%, 49.8%, 50.1% and 30.3% respectively. Moreover, the means of Scr, BUN and D-dimer were shown 6.4-folds, 1.8-folds and 0.67-folds, respectively, higher in ICU/severe cases than in corresponding non-ICU/non-severe patients. The prevalence of AKI was about 30 folds higher in ICU/severe patients compared with the non-ICU/non-severe cases.

Additional clarifications:

In addition to respond the above comments, all spelling and grammatical errors have been double checked again and corrected.

We look forward to hearing any feedback from you about our new submission and to respond any further questions and comments you may have.

Thanks again for your patience.

Sincerely,

Qixin Yang, Xiyao Yang

10/16/2020

Reference

1. Aggarwal S, Garcia-Telles N, Aggarwal G, Lavie C, Lippi G, Henry BM. Clinical features, laboratory characteristics, and outcomes of patients hospitalized with coronavirus disease 2019 (COVID-19): Early report from the United States. Diagnosis. 2020;7: 91–96. doi:10.1515/dx-2020-0046

2. Antinori S, Cossu MV, Ridolfo AL, Rech R, Bonazzetti C, Pagani G, et al. Compassionate remdesivir treatment of severe Covid-19 pneumonia in intensive care unit (ICU) and Non-ICU patients: Clinical outcome and differences in post-treatment hospitalisation status. Pharmacological Research. 2020;158: 104899. doi:10.1016/j.phrs.2020.104899

3. Cummings MJ, Baldwin MR, Abrams D, Jacobson SD, Meyer BJ, Balough EM, et al. Epidemiology, clinical course, and outcomes of critically ill adults with COVID-19 in New York City: a prospective cohort study. The Lancet. 2020;395: 1763–1770. doi:10.1016/S0140-6736(20)31189-2

4. Arentz M, Yim E, Klaff L, Lokhandwala S, Riedo FX, Chong M, et al. Characteristics and Outcomes of 21 Critically Ill Patients With COVID-19 in Washington State. JAMA. 2020;323: 1612. doi:10.1001/jama.2020.4326

5. Hirsch JS, Ng JH, Ross DW, Sharma P, Shah HH, Barnett RL, et al. Acute kidney injury in patients hospitalized with COVID-19. Kidney International. 2020;98: 209–218. doi:10.1016/j.kint.2020.05.006

6. Su H, Yang M, Wan C, Yi L-X, Tang F, Zhu H-Y, et al. Renal histopathological analysis of 26 postmortem findings of patients with COVID-19 in China. Kidney International. 2020;98: 219–227. doi:10.1016/j.kint.2020.04.003

7. Bradley BT, Maioli H, Johnston R, Chaudhry I, Fink SL, Xu H, et al. Histopathology and ultrastructural findings of fatal COVID-19 infections in Washington State: a case series. The Lancet. 2020;396: 320–332. doi:10.1016/S0140-6736(20)31305-2

8. Joannidis M, Forni LG, Klein SJ, Honore PM, Kashani K, Ostermann M, et al. Lung–kidney interactions in critically ill patients: consensus report of the Acute Disease Quality Initiative (ADQI) 21 Workgroup. Intensive Care Med. 2020;46: 654–672. doi:10.1007/s00134-019-05869-7

9. Teixeira JP, Ambruso S, Griffin BR, Faubel S. Pulmonary Consequences of Acute Kidney Injury. Semin Nephrol. 2019;39: 3–16. doi:10.1016/j.semnephrol.2018.10.001

10. Zhang J, Dong X, Cao Y, Yuan Y, Yang Y, Yan Y, et al. Clinical characteristics of 140 patients infected with SARS‐CoV‐2 in Wuhan, China. Allergy. 2020; all.14238. doi:10.1111/all.14238

11. Simonnet A, Chetboun M, Poissy J, Raverdy V, Noulette J, Duhamel A, et al. High Prevalence of Obesity in Severe Acute Respiratory Syndrome Coronavirus‐2 (SARS‐CoV‐2) Requiring Invasive Mechanical Ventilation. Obesity. 2020;28: 1195–1199. doi:10.1002/oby.22831

12. Jordan RE, Adab P, Cheng KK. Covid-19: risk factors for severe disease and death. BMJ. 2020; m1198. doi:10.1136/bmj.m1198

13. Johns Hopkins Coronavirus Resource Center. Racial data transparency: states that have released breakdowns of Covid-19 data by race. https://coronavirus.jhu.edu/data/racial-data-transparency. 

14. Ronco C, Reis T, Husain-Syed F. Management of acute kidney injury in patients with COVID-19. The Lancet Respiratory Medicine. 2020;8: 738–742. doi:10.1016/S2213-2600(20)30229-0

15. Ronco C, Reis T. Kidney involvement in COVID-19 and rationale for extracorporeal therapies. Nat Rev Nephrol. 2020;16: 308–310. doi:10.1038/s41581-020-0284-7

---

## [Editor Report · Decision Letter 1]

26 Oct 2020

Incidence and risk factors of kidney impairment on patients with COVID-19: a systematic review and meta-analysis

PONE-D-20-15148R1

Dear Dr. Yang,

We’re pleased to inform you that your manuscript has been judged scientifically suitable for publication and will be formally accepted for publication once it meets all outstanding technical requirements.

Kind regards,

Chiara Lazzeri

Academic Editor

PLOS ONE
---

## [Editor Report · Acceptance letter]

5 Nov 2020

PONE-D-20-15148R1 

Incidence and risk factors of kidney impairment on patients with COVID-19: a systematic review and meta-analysis 

Dear Dr. Yang:

I'm pleased to inform you that your manuscript has been deemed suitable for publication in PLOS ONE. Congratulations! Your manuscript is now with our production department. 

Kind regards, 

on behalf of

Dr. Chiara Lazzeri 

Academic Editor

PLOS ONE